# Prevalence of psychological distress, quality of life, and satisfaction among patients and family members following comprehensive genomic profiling testing: Protocol of the Quality of life for Cancer genomics and Advanced Therapeutics (Q-CAT) study

**Makoto Nishino[1], Maiko Fujimori [2]\*, Takafumi Koyama[1], Makoto Hirata[3], Noriko Tanabe[3], Toshio Shimizu[1], Noboru Yamamoto[1], Yosuke Uchitomi[4]**

**1** Department of Experimental Therapeutics, National Cancer Center Hospital, Tokyo, Japan, **2** Division of Supportive Care, Survivorship and Translational Research, National Cancer Center Institute for Cancer Control, Tokyo, Japan, **3** Department of Genetic Medicine and Services, National Cancer Center Hospital, Tokyo, Japan, **4** Innovation Center for Supportive, Palliative and Psychosocial Care and Department of Psycho-Oncology, National Cancer Center Hospital, and Center for Public Health Sciences, National Cancer Center, Tokyo, Japan

\* mfujimor@ncc.go.jp

## Abstract

Precision medicine is rapidly changing the diagnostic and treatment spectrum of oncology. In May 2019, comprehensive genomic profiling (CGP) (somatic and/or germline) was approved for reimbursement in Japan. While the promise of novel and targeted therapies has elevated hopes for the benefits of CGP, the lack of relevant genomic findings and/or limited access to relevant therapies remain important themes in this field. These challenges may also negatively influence the psychology of both cancer patients and their family members. However, few studies have reported longitudinal data on quality of life (QOL) with CGP. Here, we report the protocol of a prospective study, Q-CAT (QOL for Cancer genomics and Advanced Therapeutics among patients and their family members), which aims to explore the mental burden on patients and families arising from the implementation of CGP testing by collecting real-world longitudinal data using outcomes obtained with an electronic patient report, known as ePRO. This study has been registered with the Japan Registry of Clinical Trials (jRCT1030200039).

## Introduction

Despite recent dramatic advances and achievements, the need for new paradigm shifts in oncology which lead to better survival benefits and morbidity reduction is both strong and growing [1]. Comprehensive genomic profiling (CGP) and related tailored therapies, known

**Data Availability Statement:** All data in the study will be anonymised and planned to register to the Clinical Trials Registry System: Individual Case Data Repository: UMIN-ICDR (https://www.umin.ac.jp/icdr/index.html).

**Funding:** This work is supported by Japan Society for the Promotion of Science (JSPS:https://www.jsps.go.jp/english/) grant number 20K21742 and Daiwa Securities Health Foundation(https://www.daiwa-grp.jp/dsh/). The funders had and will not have a role in study design, data collection and analysis, decision to publish, or preparation of the manuscript.

**Competing interests:** Dr. Makoto Nishino reports Honoraria from AstraZeneca, Bristol-Myers Squibb, Boehringer Ingelheim Japan, Chugai, Eli Lilly, MSD, ONO, and Taiho, outside the submitted work. Dr. Takafumi Koyama reports receiving personal fees from Chugai and Sysmex, and grants from PACT Pharma outside the submitted work. Dr. Maiko Fujimori have no conflict of interest associated with the submitted work. Dr. Makoto Hirata has no conflict of interest associated with the submitted work. Dr. Noriko Tanabe has no conflict of interest associated with the submitted work. Dr. Toshio Shimizu has a consultancy/advisory role for Takeda Oncology, and has obtained research funding for his institution from Novartis, Eli Lilly, AbbVie, AstraZeneca, Eisai, Millennium-Takeda, Bristol-Myers Squibb, Incyte, Astellas Pharma, Symbio Pharmaceuticals, 3D-Medicine, Chordia Therapeutics, Five Prime, PharmaMar, and Daiichi-Sankyo, outside the submitted work; and acts as a Scientific Committee Member for Phase 1 Trials in Hong Kong under the Consortium on Harmonization of Institutional Requirements for Clinical Research (CHAIR), Hong Kong, HKSAR China. Dr. Noboru Yamamoto has a consultancy/advisory role for Eisai, Takeda Oncology, Otsuka, Boehringer Ingelheim, Cimic and Chugai; and has obtained research funding for his institution from Astellas Pharma, Chugai, Eisai, Taiho, Bristol-Myers Squibb, Pfizer, Novartis, Eli Lilly, AbbVie, Bayer, Boehringer Ingelheim, Daiichi-Sankyo, Kyowa-Hakko Kirin, Takeda, ONO, Janssen Pharma, MSD, Merck, GSK, and Sumitomo Dainippon, outside the submitted work. Dr. Yosuke Uchitomi have no conflict of interest associated with the submitted work. This does not alter our adherence to PLOS ONE policies on sharing data and materials.

as precision oncology, may provide cancer patients with such eagerly awaited new opportunities [2]. Moreover, CGP databases will benefit future clinical trials which target rare genomic alterations by providing data on distribution and allowing access to clinical trials. Two types of comprehensive genomic profiling are generally recognized, somatic and germline genomic sequencing. CGP testing can be used to identify different types of genomic variants, including significant variants that are actionable/druggable or variants of uncertain significance.

Japanese national health care system provides the population with universal health coverage. Enrollment in the system is mandatory for all Japanese and foreign residents. All tests—including CGP testing—are first examined for approval by The Pharmaceuticals and Medical Devices Agency (PMDA) and Japanese Ministry of Health, Labor, and Welfare (MHLW). The government then decides whether to provide insurance reimbursement. With reimbursement for CGP testing now approved, Japanese citizens have high expectations for this testing, despite the fact that actionable/druggable variants are only identified in 12.2% of samples [3].

Currently, there are no guidelines for healthcare professionals on how they should explain and communicate the details of CGP testing and its results to patients; how should the patient be involved regarding the decision to undergo CGP testing; or how should we deal with uncertainty in such a complex area [4]. While guidelines are being debated, little is known about the knowledge of cancer patients and their family members have about CGP, or how they feel toward it. Moreover, little is known about their values and preferences for subsequent treatment, their anxiety experiences after receiving CGP test results, or the psychological impact that these outcomes will have [5]. A very few studies have examined the responses of cancer patients recommended and underwent CGP testing with results demonstrating that patients are complaining of too much information and are misunderstanding that leads to unrealistic expectations [6–8]. Patient hopes that CGP testing would be effective was heightened with the promise of brand new genomically matched therapies only then to be challenged by no identified actionable/druggable variants, or by limitation of access to relevant genomically matched trials [6].

To our knowledge, no studies of patients undergoing CGP testing have reported longitudinal patient reported outcome data. The availability of such data would aid the understanding of patient uncertainty in the genomic testing process and the long-term psychosocial impact on patients. To remedy this, the Q-CAT (QOL for Cancer Genomics and Advanced Therapeutics among Patients and Their Family Members) study is being conducted as a psycho-social/ethical study with longitudinal data collection; at the time of consent of CGP testing (T0), at receipt of the result of CGP testing (T1), three months after receipt of the result of CGP testing (T2), six months after receipt of the result of CGP testing (T3). By following the patient longitudinally, we could evaluate both immediate and gradual reality in psychosocial influence of CGP testing toward patients and their family members.

## Objective

The aims of this study are to identify the prevalence of psychological distress, quality of life (QOL), and satisfaction with CGP testing among patients and their family members by gathering electronic patient-reported outcomes (ePROs), and qualitative data.

## Methods

### Study design

The study will be conducted under a prospective cohort design to investigate the clinical characteristics and background of cancer patients who undergo CGP and their family members.

The aims of this study in advanced cancer patients undergoing CGP and their family members are as follows:

1. Before and after the receipt of results of CGP testing (immediately, 3 and 6 months later): prevalence of depression, prevalence of anxiety, QOL, symptoms, attitude towards CGP testing, and knowledge of CGP testing.

2. Before receipt of results of CGP testing: status of social support, and medical and social background

3. After receipt of results of CGP testing: satisfaction and communication

## Setting

This cohort study will enroll patients with advanced cancer and their family members and examine data from ePROs. A total of 300 patients and about 200 family members will be recruited at the National Cancer Center Hospital, Tokyo, Japan starting May 2020. The participants will be asked to complete a survey at four time-points: before sequencing (T0), at the time of receipt of the CGP results (T1), and at three (T2) and six months after receiving the results (T3). Primary outcome is the prevalence of depression. Secondary outcomes are the prevalence of anxiety, QOL, symptoms, satisfaction, quality of communication, knowledge, state of social support, and background information, including values, attitudes, and preferences. Enabling deeper data interpretation, intentionally selected patients and their family members will be asked for permission whether to participate in four semi-structured interviews, one at each time-point. Relevant topics of QOL will be evaluated to iteratively develop or improve the support and counter-measure of QOL currently used in returning genetic results.

## Participants

Participants will be recruited at the National Cancer Center Hospital in Tokyo, Japan. The eligible patients are those with pathologically confirmed advanced or metastatic solid cancer of any histologic type that are refractory or intolerant to standard therapies, and their family members.

Patient inclusion criteria include intention to undergo CGP testing and provision of written informed consent. Patient exclusion criteria include inability to read and write Japanese, and any condition, including medical, emotional, psychiatric, or logistical condition that, in the opinion of the attending physician, would preclude the patient from adhering to the protocol or would increase the risk associated with study participation.

Inclusion criteria for family members include regular accompaniment of an enrolled patient as primary caregiver, and provision of written informed consent. Exclusion criteria for family members include inability to read or write Japanese.

## Variable

The patient-reported measures to be collected are summarized in Fig 1.

**Depression.** The nine items of PHQ-9 will be used to measure depression. Higher PHQ-9 scores indicate greater possibility of major depression (a PHQ-9 score $\geq$10, the cut off score, had a sensitivity of 88% and a specificity of 88% for major depression) [9].

**Anxiety.** The seven items of General Anxiety Disorder-7 (GAD-7) will be used to measure anxiety. Higher scores indicate greater possibility of generalized anxiety disorder (a GAD-7 score $\geq$10 had a sensitivity of 89% and specificity of 82% for generalized anxiety disorder) [10].

**Q-CAT schedule for patients**

| | | Baseline | Follow-up | | |
|---|---|---|---|---|---|
| | | T0 | T1 | T2 | T3 |
| | | Before CGP testing | At receipt of the result of CGP testing | 3 months after receipt of the result of CGP testing | 6 months after receipt of the result of CGP testing |
| Primary Endpoint | PHQ-9 | ○ | ○ | ○ | ○ |
| Secondary Endpoints | GAD-7 | ○ | ○ | ○ | ○ |
| | SCID-IV | ○ | ○ | ○ | ○ |
| | EORTC QLQ-C30 | ○ | ○ | ○ | ○ |
| | ESAS | ○ | ○ | ○ | ○ |
| | Impression of CGP testing | ○ | ○ | ○ | ○ |
| | Knowledge of CGP testing | ○ | ○ | | |
| | MSPSS | ○ | | | |
| | Medical/Social Background | ○ | | | |
| | CSQ | | ○ | | |
| | HINTS-communication | | ○ | | |

**Q-CAT schedule for family members**

| | | Baseline | Follow-up | | |
|---|---|---|---|---|---|
| | | T0 | T1 | T2 | T3 |
| | | Before CGP testing | At receipt of the result of CGP testing | 3 months after receipt of the result of CGP testing | 6 months after receipt of the result of CGP testing |
| Primary Endpoint | PHQ-9 | ○ | ○ | ○ | ○ |
| Secondary Endpoints | GAD-7 | ○ | ○ | ○ | ○ |
| | SCID-IV | ○ | ○ | ○ | ○ |
| | Impression of CGP testing | ○ | ○ | ○ | ○ |
| | Knowledge of CGP testing | ○ | ○ | | |
| | MSPSS | ○ | | | |
| | Medical/Social Background | ○ | | | |
| | CSQ | | ○ | | |
| | HINTS-communication | | ○ | | |

**Fig 1. Schedule table of Q-CAT.**

**QOL.** The 30 items of the European Organization for Research and Treatment of Cancer Quality of Life Questionnaire module Core 30 (EORTC QLQ–C30) will be used to measure QOL. Higher scores indicate a better level of global QOL scale and functional scales [11]. For symptom-oriented scales, higher scores indicate more severe symptoms.

**Qualitative interviews.** All participants with a PHQ-9 score above cut-off, and a small convenience-selected subset of participants will be invited at baseline and at each follow-up time-point to participate in structured interviews based on SCID-IV [12].

**Knowledge.** A 10-item study original questions will be used to assess knowledge on CGP testing; such as the purpose of CGP testing, availability of genomically matched therapeutic options, limitation of CGP testing, likely frequency of informative results, methodology of CGP testing. Scores will be summed from 0 to 10, with high scores indicating greater knowledge.

**Satisfaction with oncologist.** The five items of the Patient Satisfaction Questionnaire (PSQ) will be used to measure satisfaction. Lower scores indicate higher satisfaction [13].

**Quality of communication.** The 11 items on the quality of communication in the Health Information National Trends Survey (HINTS) will be used to measure quality of communication. High scores indicate greater communication [14].

**Symptoms.** The nine items of the Edmonton Symptom Assessment System (ESAS) will be used to measure symptoms. High scores indicate greater morbidity [15].

**Demographic data.** Age, gender, marital status, occupation and education level will be collected by patient report at baseline.

**Disease details.** Data available from the patient's electronic health record at the National Cancer Center Hospital, including primary site of the cancer, detailed TNM staging, cancer family history, Eastern Cooperative Oncology Group (ECOG) performance status, treatment and co-morbidities, will be collected at baseline. The results of CGP testing will be collected later, including gene alterations, whether they are actionable, whether they are druggable, whether the patients actually receive genomically matched therapy.

**Social support.**   The seven items of The Multidimensional Scale of Perceived Social Support (MSPSS) will be used to measure quality of communication. High scores indicate greater social support [16].

## Data source and procedure

Patients will provide consent for the study to the Q-CAT staff at the same time they give consent to the CGP or within one week thereafter. Subjects will be asked to fill out an ePRO survey application software or hard copy at baseline (T0), 1 to 4 weeks after receiving the results of CGP testing (T1; approximately 2–3 months post baseline), 3 months (T2) after T1, and 6 months (T3) after T1. These timelines were chosen to understand the both short- and long-term impact of the test results: short-term impact was defined at the timing of treatment decision was made, and longer-term impact was defined at the timing of undergoing or close to completion of the genomically matched therapeutics, if they are taken. Additionally at each time point, selected patients will be interviewed used the Structured Clinical Interview for DSM-V-RV (SCID), which is a semi-structured interview for making the major *DSM-5* diagnoses such as mood disorders, neurotic, stress-related and somatoform disorders, and behavioral syndromes associated with physiological disturbances and physical factors [17–19]. Patients to be interviewed include all patients with a Patient Health Questionnaire (PHQ-9) total score of > 10 and approximately 20–40 participants who score <10 as a convenience sample. The 4 week and lifetime prevalence rates of comorbid psychiatric disorders will be determined easier by this scale (Fig 2).

## Interviewer training and quality standards

Study interviewers will include two clinical psychologists and one psychiatrist who have four to 21 years clinical and research experience in psychiatry and psycho-oncology. All interviewers will undergo training before they conduct interviews using the manual of the SCID. The interviews will be performed in each combination with two of the three interviewers present, who both independently scored the answers until the kappa values will be 1.0 for all diagnosis of the interviews, indicating substantial agreement of all diagnoses between the two interviewers. In addition, each interview will be documented, and the contents will be confirmed after review by the interviewers. The gold standard will be a psychiatrist; if any discrepancies in diagnoses arise, the diagnoses will not be adopted.

## Sample size

The Q-CAT study is designed to recruit 300 patients and 192 family members over a one-year period. These sample sizes were estimated based on the fact that an average of 33 patients were tested from September to October 2019, suggesting that 396 patients will be tested over a one-year period in our institute. Given an estimated participation rate in our Q-CAT study of 75%, we determined a sample size of 300. For family members, an estimated sample size of 192 was chosen, based on an accompaniment rate by family members of patients visiting our outpatient clinic of 80%, with a consent rate of 80%.

## Statistical method and data analysis

To examine the possibility of selection bias, a comparative analysis of respondents and non-respondents will be conducted based on demographic and clinical data. Estimation of the prevalence of depression, anxiety, and other mental comorbidity in the target population will be reported as unadjusted raw rates observed in the total sample. Descriptive statistical analysis

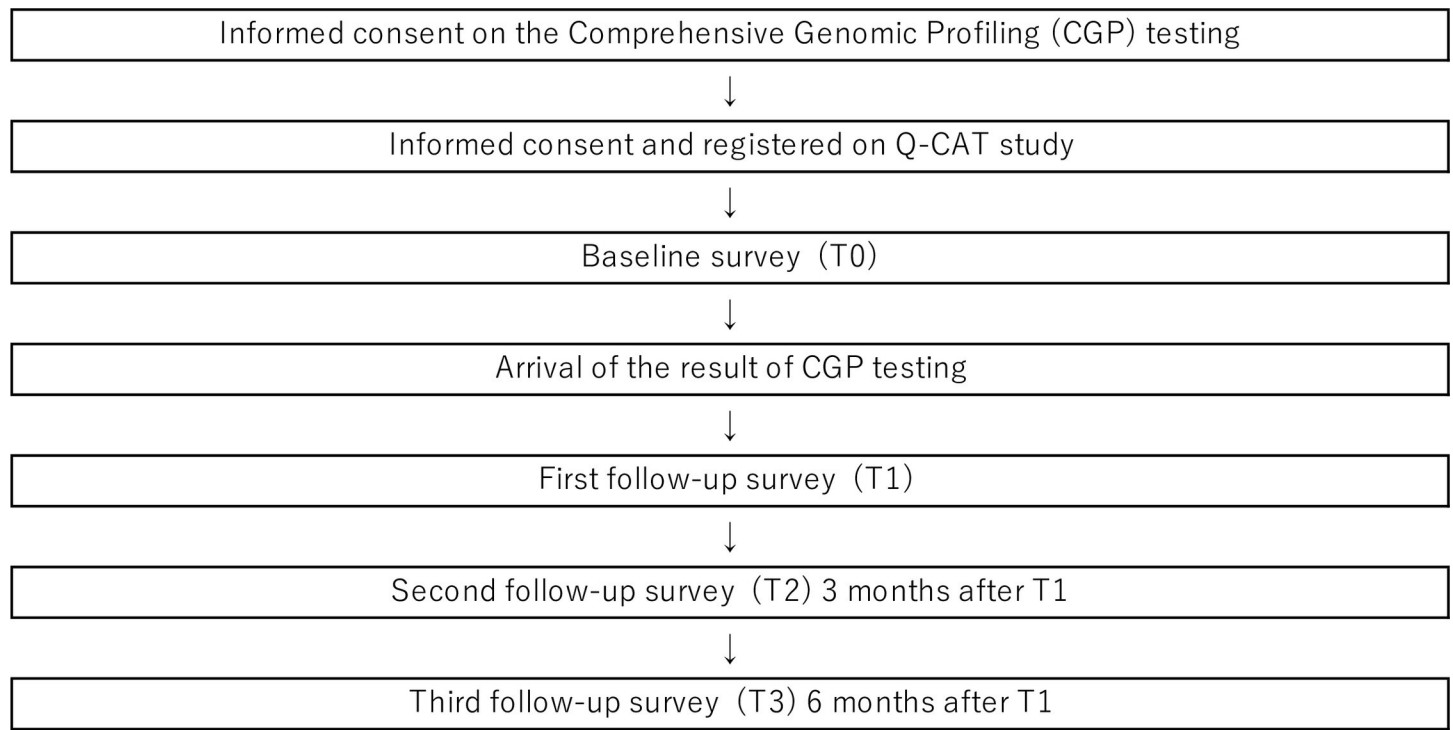

**Fig 2. Patient flow diagram.**

will be conducted for all data. By analyzing the differences between each time points, temporal changes in variables will be investigated. Bivariate associations and comparisons of the outcome scores, depression, anxiety and QOL, with variables such as knowledge, satisfaction with oncologist, quality of communication, symptoms, demographic factors, disease details, social support, and the results of CGP testing, will be conducted using correlation analysis, chi-squared tests, t-tests and ANOVA, and Kruskal-Wallis tests. In order to adjust for the effects of confounding factors, and to identify predictive factors of outcome, multivariate associations between variables will be conducted using Multiple or logistic regression analysis with depending on the outcomes. All analyses will be performed using IBM SPSS Statistics Version 26.

## Ethics and dissemination

This study has been approved by the National Cancer Center Institutional Review Board (2019–315). The results will be published in a peer-reviewed scientific academic journal, and disseminated at the nationally and internationally.

## Discussion

This study focuses on QOL among those who undergo CGP testing, using ePROs.

CGP testing has the potential to embody "precision oncology", and in turn lead to genome-related, tailored therapeutics in accordance with the genomic alteration of individual cancer patients. The true benefits of precision oncology will only be achieved if patients benefit from matched therapies through genomic alterations identified by CGP testing. Presently, however, despite dramatic growth in the expectations of both patients and family members, only a small portion of patients who undergo CGP testing—approximately 10%—obtain such benefit. To

better understand and eventually bridge this gap between perception and reality, we have planned this study to give consideration to the feelings of patients and their families.

There is also the uncertain possibility that the germline variant will be discovered, which has subsequent implications for testing the blood relatives of patients. And so, physicians who ordered and obtained consent for CGP testing often face the challenge of communicating the uncertainty both to ensure informed consent for testing, and to diminish their unrealistic expectations.

Among cancer patients who have actually undergone CGP testing, very little is known about the psychological impact of CGP testing, their knowledge of and attitudes toward CGP testing, their preferences and values for subsequent treatment, and their experience of uncertainty after receiving the test results. Patients' hopes to get benefit from CGP testing will be further enhanced by the realization of genomically matched therapies but challenged by no identified actionable/druggable variants or by limited access to genomically matched trials.

To our knowledge, no previous studies of patients underwent CGP testing have reported longitudinal patient reported outcome data which would allow an understanding of the experience of patient uncertainty across the CGP testing process and the long-term psychosocial impact on the patients. The Q-CAT study constitutes a psycho-social/ethical study to gather ePROs and qualitative data.

Study limitation includes an institutional bias, as this research is performed in a single institute; however, our institute is one of the largest high volume cancer centers in Japan with higher rates of genomically matched therapy options. Moreover, patients and/or family members may be lost to follow up (patient death, not coming to appointments, etc) given this research is collecting longitudinal ePROs data, thus results may be incomplete. Even though we didn't limit cancer types, there might be a possibility of having selection bias of patients who have higher risk cancer, thus needing GCP data. These patients may have more underlying depression and anxiety than someone who has a lower risk cancer. This would speak to the generalizability of the collected data for all patients who have cancer no matter their risk. In order to approach this issue, analysis by cancer types might be helpful to understand the backgrounds. Some elements may become confounding factors that affect the mood during the interviews, such as symptoms and other life events during the clinical course.

This will be the first study in Japan to longitudinally collect QOL data via ePRO on the experience of cancer patients and their family members with CGP. The results obtained will be analyzed according to participant background and will be used in ongoing ethical discussions, including how to effectively obtain informed consent for genomic profiling, how to explain genomic results, and how to manage patient expectations.

The findings of Q-CAT study are essential to ensuring that as CGP testing gradually integrates into a part of routine medical care, ethical considerations are taken into account, and the needs of patients and their family members are adequately supported during and after receipt of results.

Given the scarcity of longitudinal QOL data obtained via ePRO, the results of our research might become important educational tools for medical staff in providing the results of CGP testing to cancer patients and their family members. Further, the results will surely form the fundamental basis of psychological support tools for cancer patients undergoing CGP testing.

## Supporting information

**S1 Checklist. SPIRIT checklist of Q-CAT study.**
(DOC)

**S1 File. English translation protocol of Q-CAT study.**
(DOCX)

**S2 File. Japanese protocol of Q-CAT study.**
(DOCX)

## Acknowledgments

The authors are grateful to our Q-CAT colleagues for the precise and sophisticated dedication they provide every day, especially Midori Shimaoka, Ayako Sato, Risa Ohtoshi, Ikumi Tanaka, Tomo Fujiwara and Yuko Okamura. Also, we wish to express our strongest gratitude to Tsuyoshi Kawamura for creating the wonderful questionnaire application for utilizing ePRO for Q-CAT on the iPad. Q-CAT could not be realized without the commitment of the doctors, nurses, and all medical members of the National Cancer Center Hospital Tokyo, Japan; and foremost and by far, of the patients and their families themselves.

## Author Contributions

**Conceptualization:** Makoto Nishino, Maiko Fujimori, Takafumi Koyama, Makoto Hirata, Noriko Tanabe, Toshio Shimizu, Noboru Yamamoto, Yosuke Uchitomi.

**Data curation:** Makoto Nishino, Maiko Fujimori, Takafumi Koyama, Yosuke Uchitomi.

**Formal analysis:** Makoto Nishino, Maiko Fujimori, Takafumi Koyama, Toshio Shimizu, Noboru Yamamoto, Yosuke Uchitomi.

**Funding acquisition:** Maiko Fujimori, Yosuke Uchitomi.

**Investigation:** Makoto Nishino, Maiko Fujimori, Takafumi Koyama, Yosuke Uchitomi.

**Methodology:** Makoto Nishino, Maiko Fujimori, Takafumi Koyama.

**Project administration:** Makoto Nishino, Maiko Fujimori, Takafumi Koyama, Yosuke Uchitomi.

**Resources:** Makoto Nishino, Maiko Fujimori, Takafumi Koyama, Noboru Yamamoto, Yosuke Uchitomi.

**Software:** Maiko Fujimori, Yosuke Uchitomi.

**Supervision:** Maiko Fujimori, Takafumi Koyama, Makoto Hirata, Noriko Tanabe, Toshio Shimizu, Noboru Yamamoto, Yosuke Uchitomi.

**Validation:** Makoto Nishino, Maiko Fujimori, Yosuke Uchitomi.

**Visualization:** Makoto Nishino, Maiko Fujimori, Takafumi Koyama, Makoto Hirata, Noriko Tanabe, Yosuke Uchitomi.

**Writing – original draft:** Makoto Nishino, Maiko Fujimori, Takafumi Koyama, Makoto Hirata, Noriko Tanabe, Toshio Shimizu, Noboru Yamamoto, Yosuke Uchitomi.

**Writing – review & editing:** Makoto Nishino, Maiko Fujimori, Takafumi Koyama, Makoto Hirata, Noriko Tanabe, Toshio Shimizu, Noboru Yamamoto, Yosuke Uchitomi.

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
