## [Decision Letter · Decision Letter 0]

2 Nov 2022

PONE-D-22-10761Prevalence of psychological distress, quality of life, and satisfaction among patients and family members following comprehensive genomic profiling testing: Protocol of the Quality of Life for Cancer genomics and Advanced Therapeutics (Q-CAT) StudyPLOS ONE

Dear Dr. Fujimori,

Thank you for submitting your manuscript to PLOS ONE. After careful consideration, we feel that it has merit but does not fully meet PLOS ONE’s publication criteria as it currently stands. Therefore, we invite you to submit a revised version of the manuscript that addresses the points raised during the review process. Please address each reviewer comment point by point .

We look forward to receiving your revised manuscript.

Kind regards,

John W Glod

Academic Editor

PLOS ONE

“This work is supported by Japan Society for the Promotion of Science (JSPS) grant number 20K21742 and Daiwa Securities Health Foundation.”

“This work is supported by Japan Society for the Promotion of Science (JSPS:https://www.jsps.go.jp/english/) grant number 20K21742 and Daiwa Securities Health Foundation(https://www.daiwa-grp.jp/dsh/).

 The funders had and will not have a role in study design, data collection and analysis, decision to publish, or preparation of the manuscript.”

“I have read the journal's policy and the authors of this manuscript have the following competing interests:MN has reported honoraria from AstraZeneca, Bristol-Myers Squibb, Boehringer Ingelheim Japan, Chugai, Eli Lilly, MSD, Novartis, Pfizer, ONO, and Taiho, outside the submitted work.

TK has reported receiving personal fees from Chugai and Sysmex, and grants from PACT Pharma outside the submitted work.

TS has a consultancy/advisory role for Takeda Oncology, and has obtained research funding for his institution from Novartis, Eli Lilly, AbbVie, AstraZeneca, Eisai, Millennium-Takeda, Bristol-Myers Squibb, Incyte, Astellas Pharma, Symbio Pharmaceuticals, 3D-Medicine, Chordia Therapeutics, Five Prime, PharmaMar, and Daiichi-Sankyo outside the submitted work; and acts as a Scientific Committee Member for Phase 1 Trials in Hong Kong under the Consortium on Harmonization of Institutional Requirements for Clinical Research (CHAIR), Hong Kong, HKSAR China.

NY has a consultancy/advisory role with Eisai, Takeda Oncology, Otsuka, Boehringer Ingelheim, Cimic and Chugai, and has obtained research funding for his institution from Astellas Pharma, Chugai, Eisai, Taiho, Bristol-Myers Squibb, Pfizer, Novartis, Eli Lilly, AbbVie, Bayer, Boehringer Ingelheim, Daiichi-Sankyo, Kyowa-Hakko Kirin, Takeda, ONO, Janssen Pharma, MSD, Merck, GSK, and Sumitomo Dainippon, outside the submitted work.

The remaining authors declare no competing interests.”

5. We note that the original protocol file you uploaded contains a confidentiality notice indicating that the protocol may not be shared publicly or be published. Please note, however, that the PLOS Editorial Policy requires that the original protocol be published alongside your manuscript in the event of acceptance. Please note that should your paper be accepted, all content including the protocol will be published under the Creative Commons Attribution (CC BY) 4.0 license, which means that it will be freely available online, and any third party is permitted to access, download, copy, distribute, and use these materials in any way, even commercially, with proper attribution.

Therefore, we ask that you please seek permission from the study sponsor or body imposing the restriction on sharing this document to publish this protocol under CC BY 4.0 if your work is accepted. We kindly ask that you upload a formal statement signed by an institutional representative clarifying whether you will be able to comply with this policy. Additionally, please upload a clean copy of the protocol with the confidentiality notice (and any copyrighted institutional logos or signatures) removed.

Reviewers' comments:

Reviewer's Responses to Questions

**Comments to the Author**

1. Does the manuscript provide a valid rationale for the proposed study, with clearly identified and justified research questions?

Reviewer #1: Yes

Reviewer #2: Yes

2. Is the protocol technically sound and planned in a manner that will lead to a meaningful outcome and allow testing the stated hypotheses?

Reviewer #1: Yes

Reviewer #2: Yes

3. Is the methodology feasible and described in sufficient detail to allow the work to be replicable?

Reviewer #1: Yes

Reviewer #2: Yes

4. Have the authors described where all data underlying the findings will be made available when the study is complete?

Reviewer #1: Yes

Reviewer #2: Yes

5. Is the manuscript presented in an intelligible fashion and written in standard English?

Reviewer #1: Yes

Reviewer #2: No

6. Review Comments to the Author

You may also provide optional suggestions and comments to authors that they might find helpful in planning their study.

Reviewer #1: This paper addresses an important question. In my clinical practice, I have often wondered if CGP causes more satisfaction than dissatisfaction or vice versa. The study uses validated surveys to address the psychological impacts of genomic sequencing. The paper very clearly lays out the rationale, objectives and methods. The one exception, I thought, was explaining how family members are selected. I was unclear about whether they statistically analyze patients and family members seperately. If not, why is there a different target enrollment for those two populations.

Reviewer #2: Overall, this article has an interesting concept and the protocol sounds very interesting. Most comments are based on recommendations to better clarify sentences and improve readability.

Line 4-5: Sentence is confusing and reference needs to be moved before the period. Consider: “Comprehensive genomic profiling (CGP) and related tailored therapies guided by genomic findings, known as precision oncology, may provide cancer patients with early awaited new opportunities [2].”

Line 8: Suggest adding the word “allowing” so the sentence reads “and allowing access to clinical trials.”

Line 10-11: Sentence is confusing. Consider: “CGP testing can be used to identify different types of genomic variants, including significant variants that are actionable/druggable or variants of uncertain significance.”

Line 17-19: Sentences are confusing. Consider: “With reimbursement for CGP testing now approved, Japanese citizens have high expectations for this testing, despite the fact that actionable/druggable variants are only identified in 13.3% of samples [3].”

Line 20: Would recommend removing “Of note” and replace with “Currently,”

Line 23: place reference before the period. It should look like this: “complex area [4].”

Line 23-24: Suggest editing sentence for clarity: “While guidelines are being debated, very little is known on the knowledge cancer patients and their family members have about CGP or how they feel toward it.”

Line 27-30: Run-on sentence. Would recommend: “Only a handful of studies have explored the responses of cancer patients who have been offered CGP testing with results demonstrating patients reporting information overload and misunderstanding leading to unrealistic expectations, anxiety, and uncertainty [6-8].

Line 31-32: Would recommend replacing “non-findings” with “no identified actionable/druggable variants”.

Line 32: reference needs to come before the period. Should read “to relevant trials [6].”

Line 33: Consider removing “actually” for smoother reading

Line 34: Clarify what longitudinal data? I assume longitudinal patient reported outcome data.

Line 35-36: Suggest the following for clarity: “The availability of such data would aid the understanding of patient’s uncertainty across the genomic testing process…….”

Line 38: suggest removing “long span of four times” and replacing with “longitudinal data collection:”

Line 41: suggest removing “candidate for enough duration of time” with “the patient longitudinally,”

Line 44: can abbreviate quality of life as QOL and utilize in rest of paper.

Line 46: suggest introduction of abbreviation of electronic patient-reported outcomes (ePROs) in this line as is the first statement and then utilizing abbreviation in Line 54.

Line 56: Suggest replacing “from” to “starting”

Line 66-73: Suggest moving the aims of the study to under the heading: study design instead of setting.

Line 69: only need QOL, can delete “quality of life”

Line 90: Suggest adding “the” in front of Q-CAT staff

Line 91-93: Suggest removing the sentence starting with “Thus, at baseline they will……”

Line 93-94: Recommend only using ePRO as already defined earlier.

Line 99: Define PHQ-9 before using abbreviation

Line 99-106: run on sentence. Recommend editing to: “Additionally at each time point, selected patietns will be interviewed using the Structured Clinical Interview for DSM-V-RV (SCID), which includes items on mood disorders, neurotic, stress-related and somatoform disorders, and behavioral syndromes associated with physiological disturbances and physical factors [17-19]. Patients to be interviewed include all patients with a PHQ-9 total score of > 10 and approximately 20-40 participants who score <10 as a convenience sample. These assessments will facilitate the determination of the 4 week and lifetime prevalence rates of comorbid mental disorders (Fig 1).”

Line 108: replace “be” with “include”

Line 109: no apostrophe needed after years.

Line 111: recommend “after review by the interviewer.”

Line 117: PHQ-9 was utilized earlier in Line 99 and needs to be defined there with this line only using the abbreviation.

Line 117: reference needs to go at the end of the sentence.

Line 120: need reference for the sensitivity and specificity data

Line 120: Anxiety needs to be moved as the heading of the next paragraph

Line 121: Reference needs to be at the end of the sentence

Line 123: need reference for the sensitivity and specificity data

Line 126: reference needs to go at the end of the sentence

Line 141: reference needs to go before the period.

Line 145: reference needs to go before the period.

Line 147: reference needs to go at the end of the sentence.

Line 160: reference needs to go before the period

Line 190-191: can use ePROs as the abbreviation is already identified above.

Line 209: Would recommend replacing “non-findings” with “no identified actionable/druggable variants”.

Line 211: delete extra space before the start of the sentence.

Line 212: would recommend stating “longitudinal patient report outcome data”

Line 215: Replace “electronic patient-reported outcomes” with ePROs

Line 216-217: Would recommend stating: “Study limitations include an institutional bias, as this research is performed in a single institute; however, our institute is one of the largest high volume cancer centers in Japan with higher rates of genomically matched therapy options.”

Line 221: I see you do specify longitudinal QOL data in this sentence, can utilize that specific phrase in the above comments to define the longitudinal data.

Line 223: would recommend deleting “in the real world”.

Other comments:

- Interview isn’t qualitative (which would be unstructured) but rather structured and more of a diagnostic interview. Are their plans for verifying the results and how will discrepancies be addressed?

- There is no description in the statistical section of confounding elements that affect mood, etc such as pain, other life events. How will this be accomplished?

- Need to discuss other limitations such as: 1. patients and/or family members may be lost to follow up (patient death, not coming to appointments, etc) given this is longitudinal data, thus results may be incomplete. 2. Selection bias of patients who have higher risk cancer, thus needing GCP data. These patients may have more underlying depression and anxiety than someone who has a lower risk cancer. This would speak to the generalizability of the collected data for all patients who have cancer no matter their risk. 3. The interview is based on a few interviewers, which could skew how things are documented or described. 4. Other factors affect mood, etc and with longitudinal PRO data collection, other life events can affect the results from the patient.

7. PLOS authors have the option to publish the peer review history of their article (what does this mean?). If published, this will include your full peer review and any attached files.

Reviewer #1: No

Reviewer #2: No

---

## [Author Response · Author response to Decision Letter 0]

16 Jan 2023

2023/Jan/16

Dear Mr. John W Glod, Academic Editor of PLOS ONE, and Reviewers.

Happy New Year

Thank you very much for patience of waiting our institutional representative to answer #5, .

The last requirement out of 6, that they couldn’t answer Dec 2022.

Our institutional representative formally replied that we don’t need any formal statement,

by modifying the content. We, principal investigator, corresponding author and first author, agree our protocol to be shared publicly and be published. We accept your policy. The study sponsor and our institute don’t impose any restriction on sharing our documents. 

We would appreciate it if you kindly teach us how to overcome current issue.

As of additional 2 comments you gave us on Jan 6th,

We note that several of your files are duplicated on your submission. Please remove any unnecessary or old files from your revision, and make sure that only those relevant to the current version of the manuscript are included.

We removed the old files.

Please amend your list of authors on the manuscript to ensure that each author is linked to an affiliation.

We note that you have included affiliation numbers 1,2,3 and 4 however only affiliations 1,2 and 3 have authors linked to them. Please amend affiliation 4 to link an author to it or remove if added in error.

We amend affiliation 4 to link our author.

Thank you very much for your kind considerations.

We are looking forward to hearing from you.

Truly yours, Maiko

2022/Dec/16

Dear Mr. John W Glod, Academic Editor of PLOS ONE, and Reviewers.

Thank you very much for giving us a chance for revision.

As you gave us 6 requirements, we answered separately one by one as follows; except #5.

Now we are asking and waiting our institutional representative for formal statement to answer #5, 

which takes more time than we had expected.

We would appreciate it if you kindly allow us some more time for receiving formal statement with signs in order to complete our final step of revision.

All reaction in cover letter and Revised Manuscript with Track Changes is highlighted with yellow.

#1 We modified the style requirements for PLOS ONE’s style.

#2 We removed funding-related text from the manuscript.

We amended funding-related statements within our cover letter.

#3 We amended COI statements within our cover letter.

#4 We amended Data Availability statement within our cover letter

#5 Ongoing

#6 We include captions for your Supporting Information files at the end of our manuscript.

Thank you very much for your kind considerations.

We are looking forward to hearing from you.

Truly yours, Maiko

---

## [Decision Letter · Decision Letter 1]

21 Mar 2023

Prevalence of psychological distress, quality of life, and satisfaction among patients and family members following comprehensive genomic profiling testing: Protocol of the Quality of Life for Cancer genomics and Advanced Therapeutics (Q-CAT) Study

PONE-D-22-10761R1

Dear Dr. Fujimori,

We’re pleased to inform you that your manuscript has been judged scientifically suitable for publication and will be formally accepted for publication once it meets all outstanding technical requirements.

Kind regards,

John W Glod

Academic Editor

PLOS ONE

Additional Editor Comments (optional):

Reviewers' comments:

Reviewer's Responses to Questions

**Comments to the Author**

1. Does the manuscript provide a valid rationale for the proposed study, with clearly identified and justified research questions?

Reviewer #2: Yes

2. Is the protocol technically sound and planned in a manner that will lead to a meaningful outcome and allow testing the stated hypotheses?

Reviewer #2: Yes

3. Is the methodology feasible and described in sufficient detail to allow the work to be replicable?

Reviewer #2: Yes

4. Have the authors described where all data underlying the findings will be made available when the study is complete?

Reviewer #2: Yes

5. Is the manuscript presented in an intelligible fashion and written in standard English?

Reviewer #2: Yes

6. Review Comments to the Author

You may also provide optional suggestions and comments to authors that they might find helpful in planning their study.

Reviewer #2: Revision done very well and is now more comprehensive and readable. Very interesting study and I look forward to the results.

7. PLOS authors have the option to publish the peer review history of their article (what does this mean?). If published, this will include your full peer review and any attached files.

Reviewer #2: No

---

## [Editor Report · Acceptance letter]

16 May 2023

PONE-D-22-10761R1 

Prevalence of psychological distress, quality of life, and satisfaction among patients and family members following comprehensive genomic profiling testing: Protocol of the Quality of Life for Cancer genomics and Advanced Therapeutics (Q-CAT) Study 

Dear Dr. Fujimori:

I'm pleased to inform you that your manuscript has been deemed suitable for publication in PLOS ONE. Congratulations! Your manuscript is now with our production department. 

Kind regards, 

on behalf of

Dr. John W Glod 

Academic Editor

PLOS ONE